# Development and Validation of the Oxidative Stress Related lncRNAs for Prognosis in Esophageal Squamous Cell Carcinoma

**DOI:** 10.3390/cancers15174399

**Published:** 2023-09-02

**Authors:** Xuan Zheng, Wei Liu, Yingze Zhu, Wenyue Kong, Xin Su, Lanxiang Huang, Yishuang Cui, Guogui Sun

**Affiliations:** 1School of Public Health, North China University of Science and Technology, Tangshan 063200, China; ts19921214@163.com (X.Z.); 15732036499@163.com (Y.C.); 2School of Clinical Medicine, North China University of Science and Technology, Tangshan 063200, China; iweiweil@163.com (W.L.); yingze_zhu666@163.com (Y.Z.); wenyue_1919@163.com (W.K.); 15175600868@163.com (X.S.); hlx131718@sina.com (L.H.); 3Department of Hebei Key Laboratory of Medical-Industrial Integration Precision Medicine, Tangshan 063000, China; 4Affiliated Hospital of North China University of Science and Technology, Tangshan 063000, China

**Keywords:** ESCC, oxidative stress, lncRNA, overall survival, prognosis

## Abstract

**Simple Summary:**

Esophageal squamous cell cancer (ESCC) is an aggressive disease associated with a poor prognosis. The oxidative stress-related long non-coding RNAs (lncRNAs) play crucial roles in tumor prognosis. Our study identified seven oxidative stress-related DElncRNAs in the ESCC and build a prognostic risk model. The model exhibited an excellent ability for the prediction of overall survival (OS) and other clinicopathological traits using Kaplan–Meier (K-M) survival curves, receiver op-erating characteristic (ROC) curves, and the Wilcoxon test. Additionally, analysis of infiltrated immune cells and immune checkpoints indicated differences in immune status between the two risk groups. Finally, the in vitro experiments showed that PCDH9-AS1 overexpression inhibited proliferation ability and promoted apoptosis and oxidative stress levels in ESCC cells. In conclusion, the prognostic model constructed by oxidative stress-related DElncRNAs showed good performance in predicting the prognosis of patients with ESCC and was of great significance to guide the individualized treatment of these patients.

**Abstract:**

Esophageal squamous cell cancer (ESCC) is an aggressive disease associated with a poor prognosis. Long non-coding RNAs (lncRNAs) and oxidative stress play crucial roles in tumor progression. We aimed to identify an oxidative stress-related lncRNA signature that could predict the prognosis in ESCC. In the GSE53625 dataset, we identified 332 differentially expressed lncRNAs (DElncRNAs) between ESCC and control samples, out of which 174 were oxidative stress-related DElncRNAs. Subsequently, seven oxidative stress-related DElncRNAs (CCR5AS, LINC01749, PCDH9-AS1, TMEM220-AS1, KCNMA1-AS1, SNHG1, LINC01672) were selected based on univariate and LASSO Cox to build a prognostic risk model, and their expression was detected by RT-qPCR. The model exhibited an excellent ability for the prediction of overall survival (OS) and other clinicopathological traits using Kaplan–Meier (K-M) survival curves, receiver operating characteristic (ROC) curves, and the Wilcoxon test. Additionally, analysis of infiltrated immune cells and immune checkpoints indicated differences in immune status between the two risk groups. Finally, the in vitro experiments showed that PCDH9-AS1 overexpression inhibited proliferation ability and promoted apoptosis and oxidative stress levels in ESCC cells. In conclusion, our study demonstrated that a novel oxidative stress-related DElncRNA prognostic model performed favorably in predicting ESCC patient prognosis and benefits personalized clinical applications.

## 1. Introduction

Globally, esophageal cancer (EC) is recognized as one of the most common and lethal malignant tumors. The latest statistics from GLOBOCAN 2020 reveal a staggering number of cases, with 604,000 new diagnoses and 544,000 deaths in the same year. According to these figures, EC ranks as the seventh most commonly diagnosed cancer and the sixth leading cause of cancer-related deaths [1]. Based on its pathological characteristics, EC is typically categorized into two main types: esophageal squamous cell carcinoma (ESCC) and esophageal adenocarcinoma (EAC). ESCC alone accounts for more than 85% of all cases [2]. Despite the availability of multidisciplinary treatments like surgery, chemotherapy, and radiotherapy [3], the overall effectiveness of current treatment is significantly impacted by the insidious onset and aggressive nature of ESCC, as well as the absence of highly sensitive markers, all of which contribute to a grim long-term prognosis [4]. Therefore, there is an urgent need to identify effective screening methods and develop risk stratification strategies to improve patient prognosis.

Long non-coding RNAs (lncRNAs) are a class of non-coding RNAs that exceed 200 base pairs in length. They play crucial roles in DNA transcription, RNA processing, protein synthesis, and the regulation of RNA/protein modification [5,6,7,8]. Furthermore, lncRNAs can function as competitive endogenous RNAs and protein scaffolds [9,10]. Mounting evidence indicates that lncRNAs exhibit abnormal expression patterns in various malignant tumors, and they are closely associated with tumor proliferation, cell apoptosis, invasion, metastasis, angiogenesis, genomic instability, and immune evasion [11,12,13,14]. Notably, lncRNAs are abundantly present, structurally stable, temporally and spatially specific, and cell-specific. Moreover, they can be detected in diverse human body fluids. These characteristics position lncRNAs as potential disease biomarkers with broad applications in early cancer diagnosis, monitoring treatment efficacy, and predicting disease recurrence [15].

Oxidative stress refers to an imbalance between oxidation and the antioxidant system, resulting from the accumulation of free radicals triggered by internal and external stimuli. This imbalance leads to oxidative damage in tissues and organs [16]. Oxidative stress has been implicated in various diseases, including Alzheimer’s disease, chronic obstructive pulmonary disease, cardiovascular disease, and malignant tumors [17,18,19,20]. Growing evidence suggests that oxidative stress influences the expression of numerous lncRNAs during carcinogenesis and that lncRNAs, in turn, modulate oxidative stress by either enhancing or inhibiting the oxidation/antioxidant system [21]. Therefore, the objective of this study is to identify and validate a novel prognostic signature of oxidative stress-related lncRNAs to enhance the prediction of prognosis in patients with ESCC.

In this study, our aim is to identify potentially differentially expressed oxidative stress-related lncRNAs in patients with ESCC compared to control patients. Subsequently, we constructed a novel prognostic risk model utilizing these oxidative stress-related lncRNAs, which can provide valuable prognostic information for patients with ESCC.

## 2. Materials and Methods

### 2.1. Data Source

The GSE53625 dataset [22,23] including 179 cancer tissue samples and 179 adjacent normal tissue samples was acquired from the Gene Expression Omnibus (GEO) based on the platform, date type, number of samples, the type of tissue, etc. The dataset also provided comprehensive clinical information for 179 ESCC patients, which is instrumental for prognostic validation analysis. To identify relevant oxidative stress-related genes, we retrieved a total of 436 genes from the Gene Ontology (GO) database under the category “GOBP_RESPONSE_TO_ OXIDATIVE_STRESS” (Go: 0006979).

### 2.2. Identification of DElncRNAs

The identification of differentially expressed lncRNAs (DElncRNAs) was conducted using the R package “Limma” [24]. We set the cutoffs as adj.*p*.val < 0.05 and |log_2_fold change| > 1. To visualize the results, a heatmap was generated using the “pheatmap” package, while a volcano plot was created using the “ggplot2” package in R. Furthermore, a correlation test was performed using the oxidative stress expression matrix and the DElncRNAs’ expression matrix. The screening of oxidative stress-related DElncRNAs was based on the following criteria: absolute correlation coefficient (cor) > 0.7 and a *p*-value (*p*) < 0.05.

### 2.3. Establishment and Validation of the Prognostic Model Using Oxidative Stress-Related DElncRNAs

Oxidative stress-related DElncRNAs linked to the overall survival (OS) outcome were discerned through the application of univariate Cox regression analysis. This was succeeded by the formation of a risk model utilizing Lasso regression analysis, yielding the ensuing equation: risk score = ∑(Coefi × Expi). The division of the GSE53625 dataset occurred in a manner that allocated it into distinct subsets, comprising a training set and a test set, with a ratio distribution of 7:3, correspondingly. Patients in both the training and test cohorts were categorized into high-risk and low-risk groups, employing the median risk score as the threshold. The divergence in survival prognosis between these two groups was assessed using the Kaplan–Meier (K-M) survival curve, utilizing the R packages “survival” and “survminer”. Additionally, receiver operating characteristic (ROC) curves were generated to depict the predictive performance concerning the 1-year, 3-year, and 5-year OS using the R package “survivalROC”. Moreover, for ESCC samples in the GSE53625 training cohort, the Wilcoxon rank-sum test was employed to conduct a comparative analysis of risk scores across various clinicopathological parameters.

### 2.4. Establishment and Validation of the Nomogram for the Prognostic Model

Univariate and multivariate Cox regression analyses were performed to identify independent prognostic indicators for ESCC in the training cohorts. The results of these analyses were used to establish a nomogram for predicting the 1-year, 3-year, and 5-year OS of ESCC patients. To evaluate the prognostic value of the nomogram, calibration curves were plotted.

### 2.5. Gene Set Variation Analysis (GSVA) Analysis

The GSVA method was employed to estimate the variation in gene set enrichment based on the expression data from the samples. Specifically, the “GSVA” R package was utilized to identify the functions and pathways associated with the high-risk and low-risk groups in the training cohorts. GSVA was performed to explore the enrichment of Gene Ontology and Kyoto Encyclopedia of Genes and Genomes (KEGG) pathways in the high-risk and low-risk groups. The selected reference gene sets, including c5.go.bp.v7.4.symbols.gmt (biological process, BP), c5.go.cc.v7.4.symbols.gmt (cellular component, CC), c5.go.mf.v7.4.symbols.gmt (molecular function, MF), and c2.cp.kegg.v7.4.symbols.gmt (KEGG), were downloaded from the Molecular Signature Database (MSigDB). A *p*-value adjusted for multiple testing (adj.*p*.val) threshold of less than 0.05 was considered statistically significant.

### 2.6. Evaluation of Immune Infiltration and Immune Checkpoint Gene Expression between the High-Risk and Low-Risk Groups

The degree of immune cell infiltration in the different samples was analyzed using the CIBERSORT algorithm [25], and the results were visualized with a heatmap. Subsequently, a comparative analysis was undertaken within the training cohorts to ascertain dissimilarities in the levels of immune cell types’ infiltration and the expression magnitudes of immune checkpoint genes between the high-risk and low-risk groups. The assessment was performed employing the Wilcoxon test, and statistical significance was established with a threshold of *p*-value < 0.05.

### 2.7. Tissue Samples and ESCC Cell Lines

A total of 10 pairs of tissue samples, consisting of carcinoma tissue and paracancerous tissue of ESCC, were obtained from the Pathology Department of Tangshan People’s Hospital. The paracancerous tissues, located more than 5.0 cm away from the tumor, were confirmed as normal controls through H&E stains. All procedures were conducted with the approval of the Hospital Ethics Committee (Approval No.: RMYY-LLKS-2023-097). Additionally, ESCC tissue microarrays (HEsoS160CS01) containing approximately 80 pairs of tissue samples were purchased from Outdo Biotech (Shanghai, China).

The human normal esophageal epithelial cell line Het-1A and ESCC cell lines (KYSE-30, KYSE-150, KYSE-410, TE-1, and Eca-109) were obtained from Meisen Cell (Hangzhou, China). Het-1A cells were cultured in Endothelial Cell Medium (1001 + 0025 + 1052 + 0503, ScienCell, San Diego, CA, USA), while KYSE-30 and KYSE-150 cell lines were cultured in Roswell Park Memorial Institute-1640 Medium (21870076, Gibco, Carlsbad, CA, USA) supplemented with 10% Fetal Bovine Serum (10099-141, Gibco) and 1% Penicillin-Streptomycin (15140-122, Gibco). For passaging or subculturing, parietal cells were digested using 0.25% Trypsin-EDTA (25200-056, Gibco). All cell cultures were incubated in a cell incubator at 37 °C with an atmosphere containing 5% CO_2_.

### 2.8. RNA Extraction and Reverse Transcription Quantitative Polymerase Chain Reaction (RT-qPCR)

The total RNA was isolated from paraffin-embedded tissues and cells using the Paraffin Embedded Tissue Total RNA Extraction kit (R310, GeneBetter, Beijing, China) and Trizol reagent (R011, GeneBetter), adhering to the guidelines provided by the manufacturer. The quantification and assessment of RNA concentration and purity were conducted using the SpectraMax QuickDrop (Molecular Devices, Sunnyvale, CA, USA). Subsequently, the RNA samples were reverse transcribed to synthesize complementary DNA (cDNA), using the PrimeScript™ RT Master Mix (RR036A, TaKaRa, Otsu, Japan) according to the provided protocol. Real-time PCR was performed on the QuantStudio 3 System (ThermoFisher Scientific, Waltham, MA, USA) using the TB Green^®^ Premix Ex Taq™ II (RR820A, TaKaRa). For the genes CCR5AS, LINC01749, PCDH9-AS1, TMEM220-AS1, KCNMA1-AS1, SNHG1, and LINC01672, qPCR protocols were executed. These protocols encompassed an initial denaturation step at 95 °C for 30 s, proceeded by 40 cycles of denaturation at 95 °C for 5 s, and subsequently, an annealing/extension stage at 60 °C for 34 s. The computation of relative gene expressions was carried out using the 2^−ΔΔct^ method [26], while the normalization was conducted in relation to the GAPDH gene. The primer sequences for all the indicated genes are listed in Table 1.

### 2.9. Fluorescence In Situ Hybridization (FISH) Assay

FISH analysis for PCDH9-AS1 expression was performed on the ESCC tissue microarray using the FISH kit (C10910, Ribobio, Guangzhou, China). The FISH probe specific to PCDH9-AS1, labeled with Cy3, was designed and synthesized by RiboBio (Guangzhou, China). The procedure involved deparaffinization and pretreatment of the tissue microarray, followed by protease digestion. The tissue sections were then washed and pre-hybridization buffer was added to the unstained tissue sections for 30 min. The PCDH9-AS1 FISH probe was hybridized in the dark overnight at 37 °C. Nucleus staining was carried out using DAPI (H-1200-10, Vector Laboratories, Burlingame, CA, USA) at room temperature for 5 min. All images were captured and observed under an IX81 fluorescence microscope (Olympus, Tokyo, Japan).

### 2.10. Cell Transfection

PCDH9-AS1 was cloned into the pcDNA3.1 (+) vector (V79020, Invitrogen, Carlsbad, CA, USA) by XIEBHCbio (Beijing, China). Cells were seeded onto 6-well plates and allowed to reach approximately 80% confluence. Transfection was performed using Lipofectamine^®^ 3000 (L3000015, Invitrogen) according to the manufacturer’s instructions.

### 2.11. Cell Viability Assay

KYSE-30 and KYSE-150 cells, transfected as described previously, were seeded into separate wells of a 96-well cell culture dish at a density of 2 × 10^3^ cells/well. Cell viability was assessed using the Cell Counting Kit-8 (CCK-8) (MF128-01, Mer5bio, Beijing, China). The CCK-8 reagent was added to the cell supernatant at the indicated time points and incubated with the cells at 37 °C for 2 h. The absorbance of the cells was then measured at a wavelength of 450 nm using a microplate reader (ThermoFisher Scientific, USA). 

### 2.12. Colony Formation Assay

KYSE-30 and KYSE-150 cells, transfected as described previously, were seeded into individual wells of 6-well plates at a density of 2 × 10^3^ cells per well and cultured for 7–10 days. After the incubation period, the cells were fixed with methanol for 15 min and stained with 0.1% (*v*/*v*) crystal violet (G1063, Solarbio, Beijing, China) for 15 min. The colonies containing more than 50 cells were counted, and the average number of colonies was used to evaluate the cell’s ability to form colonies, which is an indirect measure of colony-forming capability.

### 2.13. Flow Cytometry Analysis

At 48 h post-transfection, KYSE-30 and KYSE-150 cells were gathered using 0.25% trypsin and collected by centrifugation at 1000× *g* for a duration of 5 min. The staining procedure involving Annexin V and PI staining was executed in accordance with the manufacturer’s guidelines, utilizing the Annexin V-FITC apoptosis detection kit (BD Biosciences, San Jose, CA, USA). Subsequently, the cells were subjected to flow cytometric analysis using the BD FACSAria™ II instrument (BD Biosciences, USA).

### 2.14. Reactive Oxygen Species (ROS) Detection

The intracellular ROS level was measured using the Reactive Oxygen Species (ROS) assay kit (S0033S, Beyotime, Shanghai, China) following the manufacturer’s instructions. Briefly, the ROS probe H2DCFH-DA was diluted to a concentration of 10 μM using serum-free medium. The previously transfected KYSE-30 and KYSE-150 cells were incubated with 1 mL of DCFH-DA working solution in each well, in the dark, for 20 min. Subsequently, the cells were washed three times with serum-free medium to remove any residual DCFH-DA that did not enter the cells. Finally, the cells were observed and photographed under an IX81 fluorescence microscope (Olympus, Japan).

### 2.15. Determination of Lactate Dehydrogenase (LDH) Release

LDH release levels were detected using the Lactate Dehydrogenase Assay Kit (C0016, Beyotime, Shanghai, China), following the manufacturer’s protocol. KYSE-30 and KYSE-150 cells were cultured in 96-well plates and transfected as described previously. After the appropriate incubation period, the culture supernatant was collected, and LDH release levels were analyzed by measuring the absorbance at 490 nm using a microplate reader. 

### 2.16. Statistical Analysis

The statistical analyses were conducted using R software 4.1.0 (R Core Team, Auckland, New Zealand) for Statistical Analysis and Visualization. Additionally, SPSS software version 17 (SPSS Inc., Chicago, IL, USA) was used for certain analyses. The Wilcoxon test was employed to compare the differences in the risk score between different groups. Cox regression analysis was utilized to assess the prognostic power of the prognostic features. Pairwise comparisons were performed using Student’s *t*-test. The chi-squared test was employed to analyze the differences in the proportions of clinical characteristics. A *p*-value or, if necessary, an adjusted *p*-value < 0.05 was considered statistically significant.

## 3. Results

### 3.1. Screening Results of DElncRNAs in GSE53625 Dataset

The overall research design of this study is presented in Figure 1. According to the screening criteria, a total of 332 DElncRNAs were identified in the GSE53625 dataset, comprising 137 upregulated DElncRNAs and 195 downregulated DElncRNAs. The volcano plot and heatmap of the DElncRNAs are displayed in Figure 2A,B. Subsequently, based on the expression patterns of these 332 DElncRNAs and 436 oxidative stress-related genes, 174 oxidative stress-related DElncRNAs were selected for further analysis using a correlation coefficient cutoff of |cor| > 0.7 and *p*-value < 0.05.

### 3.2. Establishment and Validation of a Risk Model

First, this study classified the included cases (*n* = 179) into training (*n* = 126) and validation (*n* = 53) sets at a 7:3 ratio. In the training set, 174 oxidative stress-related DElncRNAs were screened through preliminary univariate Cox analysis (Figure 3A). To prevent overfitting of prognostic features, LASSO regression was applied to analyze these lncRNAs, resulting in the identification of seven oxidative stress-related DElncRNAs (CCR5AS, LINC01749, PCDH9-AS1, TMEM220-AS1, KCNMA1-AS1, SNHG1, LINC01672) significantly associated with survival (Figure 3B,C). Subsequently, a prognostic oxidative stress-related risk model was established using these aforementioned lncRNAs. The risk score was calculated using the following formula: risk score = CCR5AS expression × (−0.049993331) + LINC01749 expression × (−0.000917922) + PCDH9-AS1 expression × (−0.023456133) + TMEM220-AS1 expression × (−0.068543123) + KCNMA1-AS1 expression × (0.172214585) + SNHG1 expression × (0.05726297) + LINC01672 expression × (0.850776742). In both the training set and the test set, cases were divided into high-risk and low-risk groups based on the median of the risk score. The distribution of risk scores and survival status indicated a higher proportion of deceased individuals with increasing risk scores in both the training set and the test set (Figure 3D–G). 

Furthermore, K-M survival analysis conducted on the training set demonstrated a significantly shorter OS time for patients in the high-risk group compared to those in the low-risk group (*p* < 0.01) (Figure 3H). Similarly, The K-M survival curves revealed a higher survival probability for the low-risk group in the test set (*p* < 0.05) (Figure 3I). Additionally, the AUC values obtained from ROC curve analysis were 0.65, 0.71, and 0.70 for 1-year, 3-year, and 5-year OS, respectively, in the training set (Figure 3J). The corresponding AUC values for 1-year, 3-year, and 5-year OS in the test set were 0.65, 0.66, and 0.66, respectively (Figure 3K).

### 3.3. Construction of the Nomogram

To assess the prognostic value of the constructed prognostic signature, univariate and multivariate Cox analyses were conducted. The results of the univariate analysis indicated that the risk score, N stage, and TNM stage were significant predictors of poor OS in ESCC cases (Figure 4A). Furthermore, the multivariate analysis confirmed the independence of our constructed prognostic model in predicting ESCC prognosis (Figure 4B). Based on the multivariate analysis results, a nomogram incorporating the risk score, N stage, and TNM stage was constructed to predict the 1-year, 3-year, and 5-year OS for ESCC (Figure 4C). The calibration curve demonstrated good agreement between the predicted and actual patient outcomes (Figure 4D).

### 3.4. Association of the Risk Score and Clinicopathological Traits

We conducted an analysis to assess the association between the risk score and clinical characteristics, including gender, T stage, N stage, and TNM stage. The results obtained from the Wilcoxon test demonstrated that the risk score showed significant differences among different stages (stage 2 vs. stage 3, *p* < 0.05), T stages (T3 vs. T4, *p* < 0.05), and N stages (N0 vs. N1, *p* < 0.05). However, no significant difference in the risk score was observed between genders (female vs. male, *p* > 0.05). (Figure 5). 

### 3.5. GSVA between High-Risk and Low-Risk Groups

To explore the functional and pathway differences between high-risk and low-risk groups, we performed GSVA. In terms of BP, ESCC samples in the high-risk group exhibited enrichment in “epithelial cilium movement involved in determination of left-right asymmetry” and “regulation of Wnt signaling pathway planar cell polarity pathway”. Conversely, samples in the low-risk group showed enrichment in “positive regulation of granulocyte-macrophage colony stimulating factor production” and “antimicrobial humoral immune response mediated by antimicrobial peptide” (Figure 6A). The CC analysis indicated that “prespliceosome” and “npbaf complex” were enriched in the high-risk group samples, while “cornify envelope” and “sperm plasma membrane” were enriched in the low-risk group samples (Figure 6B). In terms of MF, the high-risk group exhibited enrichment in “glycerophospholipid flippase activity” and “DNA secondary structure binding”, whereas the low-risk group showed enrichment in “interleukin 1 receptor binding” and “structural constituent of skin epidermis” (Figure 6C). Furthermore, the high-risk group demonstrated enrichment in the KEGG gene sets “mismatch repair” and “spliceosome”, whereas the low-risk group exhibited enrichment in “Olfactory transduction” and “Toll-like receptor signaling pathway” (Figure 6D).

### 3.6. Correlations with Immune Microenvironment

To investigate the capacity of our risk model to portray the immunological milieu within the tumor microenvironment (TME) of ESCC, we utilized the CIBERSORT algorithm to compute the infiltration scores of distinct immune cell populations. Subsequently, we scrutinized the variations between the high-risk and low-risk groups. Analysis using the CIBERSORT approach unveiled that the low-risk group exhibited elevated levels of immune cell infiltrations, including activated mast cells and neutrophils, compared to the high-risk group (*p* < 0.05) (Figure 7A,B). Additionally, we evaluated the expression differences of immune checkpoints between the two risk groups. The results demonstrated that immune checkpoints, such as BTNL2, CD70, CD86, CTLA4, ICOSLG, LAG3, LAIR1, NRP1, TNFRSF14, and TNFRSF8, were significantly upregulated in the high-risk group (*p* < 0.05). Conversely, the high-risk group exhibited lower expressions of CD274, CD80, HHLA2, and KIR3DL1 compared to the low-risk group (both *p* < 0.05) (Figure 7C).

### 3.7. The Oxidative Stress-Related DElncRNA Validation in ESCC

Next, we examined the expression levels of the seven oxidative stress-related DElncRNAs. We assessed their expression in various cell lines, comparing them to the human normal esophageal epithelial cell line Het-1A. As depicted in Figure 8A, LINC01749, PCDH9-AS1, TMEM220-AS1, and KCNMA1-AS1 exhibited relatively lower expression levels in ESCC cell lines (KYSE-30, KYSE-150, KYSE-410, TE-1, and Eca-109) compared to Het-1A. Conversely, SNHG1 and LINC01672 demonstrated relatively higher expression levels. However, there were no significant differences in the expression levels of CCR5AS between the two groups. To further verify the expression of these biomarkers, we used qRT-PCR to compare gene expression levels in carcinoma tissue and paired paracancerous tissue from 10 ESCC patients. The clinical samples showed similar expression trends to those found in the cell lines (Figure 8B). Except for CCR5AS, the expression levels of the other six oxidative stress-related DElncRNAs were completely consistent with the results obtained from data mining. These findings further validate the accuracy of the aforementioned bioinformatics research. Based on its higher fold change, we selected PCDH9-AS1 for subsequent functional assays.

### 3.8. Loss of PCDH9-AS1 Predicted Unfavorable Prognosis of ESCC Patients

To assess the clinical significance of PCDH9-AS1, we performed FISH experiments to examine its expression levels in tissues. The results demonstrated a significantly lower average fluorescence intensity of PCDH9-AS1 in cancer tissues compared to paired adjacent non-cancerous tissues (Figure 9). Subsequently, we divided ESCC cancer tissue into low-expression and high-expression groups based on the median value of PCDH9-AS1 expression, with 40 cases in each group. We further investigated the association between PCDH9-AS1 expression levels and clinical pathological features. The results revealed a significant correlation between PCDH9-AS1 expression and the clinical staging of ESCC patients, while no correlation was observed with gender, age, pathological grading, lymph node metastasis, or tumor staging (Table 2). These findings suggest that PCDH9-AS1 may function as a tumor suppressor, and its low expression is indicative of poor prognosis in ESCC patients. 

### 3.9. Overexpression of PCDH9-AS1 Attenuates ESCC Cell Proliferation and Promotes Apoptosis and Oxidative Stress Level

To investigate the functions of PCDH9-AS1 in ESCC, a PCDH9-AS1 pcDNA3.1(+) vector was constructed (pcDNA3.1-PCDH9-AS1), resulting in the upregulation of PCDH9-AS1 (Figure 10A). Subsequently, both the CCK8 assay and colony formation assay revealed that PCDH9-AS1 overexpression repressed cell proliferation compared to the control group (pcDNA3.1) (Figure 10B,C). Nevertheless, the flow cytometry assay demonstrated that PCDH9-AS1 overexpression significantly increased the apoptosis level of ESCC cells (Figure 10D). Considering that an increase in ROS and LDH levels is a characteristic response to oxidative stress, we further explored the impact of PCDH9-AS1 upregulation on these parameters in ESCC cells. Our results indicated that PCDH9-AS1 overexpression promoted the production of ROS (Figure 10E) and significantly elevated LDH levels, as determined by an LDH assay kit (Figure 10F).

## 4. Discussion

The survival rate for esophageal cancer patients at the 5-year mark is currently low, ranging from 10% to 30% [27]. However, early detection and timely treatment have the potential to significantly increase the 5-year survival rate to over 80% [28]. Consequently, there is a growing emphasis on the identification of robust and sensitive predictive models for early diagnosis and prognosis, including those based on ferroptosis-related genes, immune genes, fibroblast-related features, immunohistochemical features, and metabolites [29,30,31,32,33]. Persistent oxidative stress is a prominent characteristic observed throughout tumor development and has been identified as a double-edged sword in tumor progression and cancer treatment [34]. Recent studies have highlighted the involvement of oxidative stress in ESCC. For instance, a Caspase-8 variant activates Nrf2 through SQSTM1 phosphorylation, resulting in the suppression of oxidative stress levels and exerting a pro-tumorigenic effect in ESCC [35]. Depletion of IFI6 leads to mitochondrial dysfunction and endoplasmic reticulum stress, which culminate in the accumulation of reactive oxygen species and inhibition of ESCC progression [36]. Additionally, Liu et al. constructed a prognostic model pertaining to oxidative stress in ESCC based on the integration of data from the TCGA, GTEx, and GEO databases. This model aims to predict the prognosis of ESCC patients and carries considerable clinical significance [37]. Consequently, the development of oxidative stress-related biomarkers is crucial for predicting the prognosis of ESCC patients. Owing to their diverse array and intricate spatial configurations, lncRNAs possess the capacity to regulate expression at multiple levels, including transcription and translation [38]. Numerous studies have demonstrated, under long-term or short-term oxidative stress, that the expression levels of lncRNAs are dysregulated. For example, in gastric cancer, DNA damage response induced by oxidative stress can promote the binding of H3K27ac and CREBBP, thus facilitating the expression of lncRNA NORAD [39]. In cholangiocarcinoma, oxidative stress can also upregulate the expression of lncRNA H19 and lncRNA HULC [40]. On the other hand, lncRNAs can affect the levels of oxidative stress in tumor cells through various mechanisms. For instance, in hepatocellular carcinoma, LINC01134 recruits the transcription factor SP1 to the p62 promoter to activate the antioxidant pathway of p62 [41]; lncRNA GABPB1-AS1 interacts with GABPB1 to inhibit its translation, leading to decreased PRDX5 expression and ultimately impaired antioxidant capacity of cells [42]. Additionally, the crosstalk between lncRNAs and oxidative stress can modulate various cancer-related signaling pathways such as p53, NF-κB, Nrf2, AKT, EGFR, FOXO3, Keap1, PTEN, and Wnt [43,44]. In our study, we successfully established a prognostic risk signature based on seven oxidative stress-related DElncRNAs, which exhibited high accuracy in predicting OS. These seven prognostic oxidative stress-related lncRNAs are closely associated with the TME and immunotherapy in ESCC. Furthermore, we conducted preliminary investigations into the anti-cancer role of PCDH9-AS1 in ESCC and found that its overexpression may attenuate ESCC cell proliferation, promote apoptosis, and induce oxidative stress.

We initially identified differentially expressed lncRNAs between the cancer group and the normal group. Subsequently, we performed correlation analysis between these lncRNAs and oxidative stress-related genes and selected lncRNAs that exhibited associations with oxidative stress. Through univariate Cox and LASSO regression analyses, we identified seven DElncRNAs that were significantly associated with prognosis and oxidative stress: CCR5AS, LINC01749, PCDH9-AS1, TMEM220-AS1, KCNMA1-AS1, SNHG1, and LINC01672. Using these seven lncRNAs, we constructed a risk model for predicting the prognosis of ESCC patients. To enhance the prediction of ESCC prognosis, we integrated the risk score and clinical features into an integrated column chart. Among the seven lncRNAs, CCR5AS, LINC01749, PCDH9-AS1, and TMEM220-AS1 were identified as low-risk factors, while KCNMA1-AS1, SNHG1, and LINC01672 were identified as high-risk factors. Several of these lncRNAs in the risk model have been reported to play significant roles in different cancers, such as CCR5AS, LINC01749, TMEM220-AS1, KCNMA1-AS1, SNHG1, and LINC01672. Notably, PCDH9-AS1 was identified as a lncRNA associated with prognosis and oxidative stress for the first time in this study. Previous bioinformatics investigations have demonstrated the prognostic value of CCR5AS and LINC01749 in melanoma and ESCC, respectively [45,46]. In liver cancer, TMEM220-AS1 functions as a competitive endogenous RNA by regulating the miR-484/MAGI1 axis to inhibit tumor progression [47]. Additionally, studies by Ma et al. have shown that KCNMA1-AS1 is highly expressed in epithelial ovarian cancer (EOC) and is negatively correlated with prognosis. KCNMA1-AS1 promotes EOC cell proliferation and migration, inhibits apoptosis, and contributes to the occurrence and development of EOC [48]. SNHG1 is an oncogenic gene that has been validated by numerous cancer researchers. For example, SNHG1 promotes ESCC cell proliferation and invasion by regulating the Notch signaling pathway [49]. In bladder cancer, SNHG1 promotes tumor cell proliferation and inhibits apoptosis by regulating PPARγ ubiquitination [50]. LINC01672, also known as lncRNA NMR, is significantly upregulated in ESCC and serves as a key regulatory factor in promoting tumor metastasis and drug resistance [51]. Interestingly, we observed that KCNMA1-AS1 was downregulated in ESCC compared to non-tumor samples, but it still functions as a “risk” lncRNA in ESCC, potentially regulating oxidative stress in conjunction with other genes to impact tumor progression. Our findings broadly align with these studies, and the identification of these oxidative stress-related lncRNAs provides valuable insights into ESCC and the identification of potential therapeutic targets.

Next, we performed GSVA analysis on high-risk and low-risk groups. ESCC samples in the high-risk group exhibited enrichment in aspects such as “regulation of Wnt signaling pathway planar cell polarity pathway”, “DNA secondary structure binding”, “mismatch repair”, and “spliceosome”, whereas the low-risk group showed enrichment in aspects such as “positive regulation of granulocyte-macrophage colony stimulating factor production”, “an-timicrobial humoral immune response mediated by antimicrobial peptide”, “inter-leukin 1 receptor binding”, and “Toll-like receptor signaling pathway”. Consultation of the literature revealed that several pathways mentioned above are involved in tumor progression. For instance, lncRNA DGCR5 increases its stability by directly binding with SRSF1, which can regulate MCL-1 alternative splicing, thereby affecting ESCC progression [52]. LncRNA HOTAIR may promote malignant progression of liver cancer by downregulating SETD2 expression and phosphorylation levels, leading to the suppression of DNA damage repair [53]. Moreover, Rebernick et al. indicated a negative correlation between the high expression of GM-CSF and survival rates in esophageal cancer patients [54]. Su et al. found that ESCC patients with high TLR3 expression have longer overall survival and their high expression is associated with immune cell infiltration and activation of apoptotic pathways [55]. Thus, we inferred that the prognostic lncRNAs obtained in this study may participate in the regulation of ESCC through mechanisms involving alternative splicing, DNA mismatch repair, GM-CSF production, the Toll-like receptor signaling pathway, and more. This provides a new theoretical foundation for further investigating the potential molecular mechanisms of these genes in ESCC. An increasing body of research suggests a strong correlation between oxidative stress and tumor immunity. For instance, Zhao et al. demonstrated that PRAK plays a vital role in regulating antioxidant stress in Th17 cells, and its intervention significantly improves the glycolytic metabolism of Th17 cells, thereby enhancing the anti-tumor immune response mediated by Th17 cells [56]. Furthermore, CD4+ T cells have been shown to modulate tumor metabolism, leading to increased TNF-α-dependent oxidative stress and subsequent tumor cell death [57]. However, limited research has been conducted on the direct relationship between oxidative stress and immune cell infiltration in ESCC. In this study, we divided the GSE53625 dataset into training and testing sets. Using the median risk scores derived from the training set as a threshold, we classified both the training set and test set samples into high-risk and low-risk groups. We observed a significant decrease in the infiltration of activated macrophages and neutrophils in the high-risk group compared to the low-risk group. Interestingly, Zhuge et al. demonstrated a positive correlation between high densities of neutrophils, macrophages, and dendritic cells and improved survival in ESCC patients [58]. Additionally, Liu et al. divided the ESCC patients into two groups and reported a higher median survival time in the first group. Similarly, the infiltration levels of neutrophils, plasma cells, and activated macrophages were higher in the first group compared to the second group, indicating a positive correlation between the infiltration levels of these cells and patient survival, which is consistent with our findings [37]. However, conflicting perspectives exist among scholars. Luo et al. indicated that increased neutrophil infiltration within the tumor is an independent adverse prognostic factor in ESCC [59]. Mao et al. found that the expression level of MMP12 positively correlates with the infiltration levels of activated macrophages and M0 macrophages, and high MMP12 expression is significantly associated with poor prognosis in ESCC patients [60]. These discrepancies may be attributed to the phenotypic and functional plasticity of neutrophils and macrophages. Cellular factors and epigenetic signals within the tumor microenvironment can induce the polarization of neutrophils into anti-tumor N1-type tumor-associated neutrophils or pro-tumor N2-type tumor-associated neutrophils [61]. Activated mast cells can release various active factors, chemokines, and cytokines that exert diverse functional roles in tumor development. Factors such as IL-8, VEGF, PDGF, NGF, SCF, and histamine promote tumor growth, while IL-1, IL-6, TNF-α, and fibroblast proteases inhibit tumor growth [62]. Therefore, we speculate that the seven oxidative stress-related lncRNAs identified in this study may influence the polarization and function of various cells in the tumor immune microenvironment, thereby impacting the prognosis of ESCC patients through mechanisms such as inducing antibody-dependent cellular cytotoxicity, exerting direct cytotoxic effects, activating anti-tumor adaptive immunity, and secreting tumor-suppressive factors. Furthermore, compared to the low-risk group, the high-risk group exhibited higher expression of immune checkpoint genes, including BTNL2, CD70, CD86, CTLA4, ICOSLG, LAG3, LAIR1, NRP1, TNFRSF14, and TNFRSF8. However, further research is required to determine whether inhibitors targeting these checkpoints can serve as promising anti-tumor drugs for ESCC.

In this study, we constructed a prognostic model based on seven lncRNAs associated with ESCC patient prognosis. By computing the patients’ risk scores, we predict their prognoses, wherein patients in the high-risk group exhibit poorer outcomes. Prognostic models hold the potential to anticipate the trajectory of patients’ future developments, enhancing comprehension of disease progression. Furthermore, we performed an independent prognostic analysis and developed a nomogram model grounded in the risk score and clinical features of the samples. This model transforms risk score, N stage, and TNM stage into scores to predict patient survival rates at 1, 3, and 5 years, offering clinical utility. Similarly, immune infiltration-related analyses also provide valuable insights for incorporating immune therapy into the treatment of ESCC patients and assessing immunotherapeutic efficacy. In summary, the oxidative stress-related prognostic model we constructed for ESCC patients using bioinformatics methods presents novel directions and targets for ESCC treatment and prognosis enhancement. However, the effectiveness and clinical applicability of this model necessitate further validation. Furthermore, we conducted an assessment of the expression levels of the seven identified oxidative stress-related DElncRNAs in both cells and tissues, and their expression patterns corresponded with the predictions generated by the previous bioinformatics analysis. Notably, the functional phenotypic analysis of PCDH9-AS1 revealed a significant suppression of cancer cell proliferation when PCDH9-AS1 was overexpressed. Moreover, it was found to enhance apoptosis and elevate oxidative stress levels. These findings suggest that PCDH9-AS1 acts as a positive factor for the prognosis of ESCC patients and exerts regulatory effects on tumor development through mechanisms associated with oxidative stress.

Nevertheless, it is important to acknowledge the limitations of this study. Firstly, the risk model for oxidative stress-related DElncRNAs was constructed using data solely obtained from the GEO database, which may introduce potential bias. To ensure the robustness and generalizability of the findings, it is necessary to validate the findings using external datasets and larger clinical cohorts. Secondly, although we conducted RT-qPCR experiments to assess the expression levels of all seven lncRNAs in 10 clinical samples and five ESCC cell lines, the sample size remains relatively small. Therefore, future studies should expand the sample size to enhance the statistical power and reliability of the results. Lastly, the underlying mechanisms by which these lncRNAs influence oxidative stress and how oxidative stress impacts the progression of ESCC remain unclear. Further investigations are needed to elucidate the intricate relationship between lncRNAs, oxidative stress, and ESCC.

## 5. Conclusions

This study aimed to identify seven DElncRNAs (CCR5AS, LINC01749, PCDH9-AS1, TMEM220-AS1, KCNMA1-AS1, SNHG1, and LINC01672) associated with oxidative stress and develop a risk model and nomogram for accurate prognosis prediction in patients with ESCC, which offers novel approaches for predicting and improving the prognosis of ESCC patients, thereby providing new avenues for ESCC treatment. Additionally, the study investigated the relationship between the risk model and the immune environment. Immune-related analyses, particularly those involving immune checkpoints, furnish theoretical support for the application of immune therapy. Concurrently, novel insights were gained into the role and potential mechanisms of a specific long non-coding RNA, PCDH9-AS1, in regulating the development of ESCC. Our research findings establish a new theoretical foundation for exploring the roles of lncRNAs and oxidative stress in ESCC, presenting fresh targets for ESCC treatment. Moreover, we will persist in investigating the roles of these prognostically relevant lncRNAs, delving into their potential molecular mechanisms. As our study unfolds, we anticipate gaining a deeper understanding of these mechanisms and their implications. In conclusion, these findings have important implications for prognosis prediction, diagnosis, and treatment of esophageal squamous cell carcinoma in clinical practice.

## Figures and Tables

**Figure 1 cancers-15-04399-f001:**
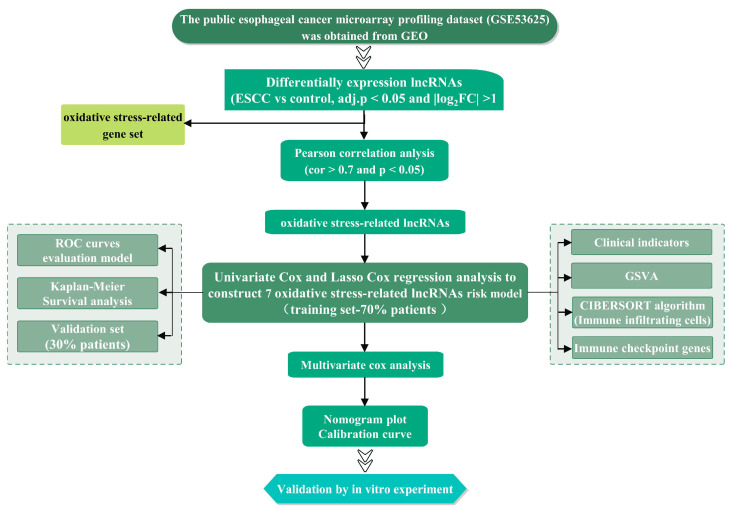
Study flowchart.

**Figure 2 cancers-15-04399-f002:**
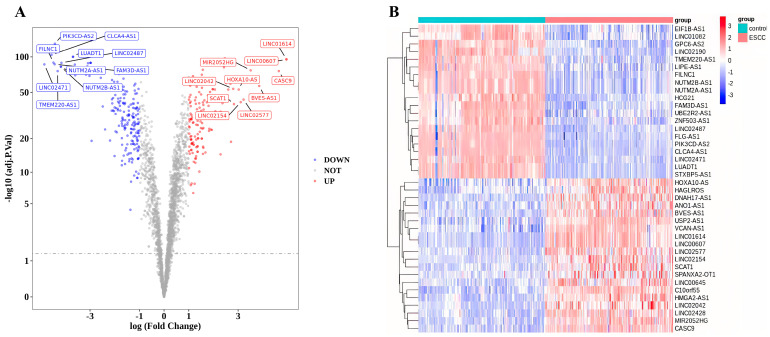
Identification of differentially expressed lncRNAs (DElncRNAs) in esophageal squamous cell carcinoma (ESCC). (**A**) Volcano plot illustrating the DElncRNAs in ESCC tissues compared with paired paracancerous tissues from the GSE53625 dataset. Red dots denote upregulated genes, blue dots represent downregulated genes, and grey dots indicate genes with no significant difference. (**B**) Heat map displaying the top 10 DElncRNAs in the GSE53625 dataset. Red represents high expression, while blue indicates low expression.

**Figure 3 cancers-15-04399-f003:**
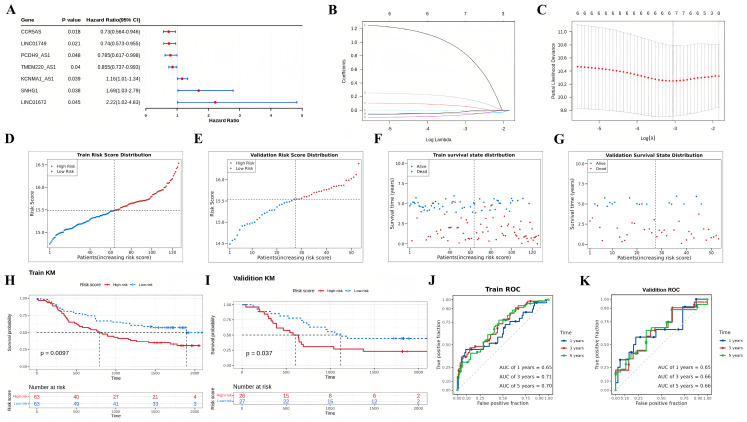
Construction and validation of a prognostic oxidative stress-related risk model composed of seven DElncRNAs. (**A**) Forest plots presenting the results of Cox univariate regression analysis of prognostic oxidative stress-related DElncRNAs. (**B**,**C**) cvfit and lambda curves illustrate the application of the least absolute shrinkage and selection operator (LASSO) regression using the minimum criteria. Each line represents one oxidative stress-related DElncRNAs (CCR5AS, LINC01749, PCDH9-AS1, TMEM220-AS1, KCNMA1-AS1, SNHG1, LINC01672) in subfigure B. (**D**–**G**) Distribution of the risk scores, overall survival status (OS), and risk score in the training and validation datasets. (**H**,**I**) Kaplan–Meier curves demonstrating the survival status and survival time in the training dataset and validation dataset. (**J**,**K**) Receiver operating characteristic (ROC) curve showcasing the potential of the prognostic oxidative stress-related DElncRNA signature in predicting 1-year, 2-year, and 3-year OS in the training dataset and validation dataset.

**Figure 4 cancers-15-04399-f004:**
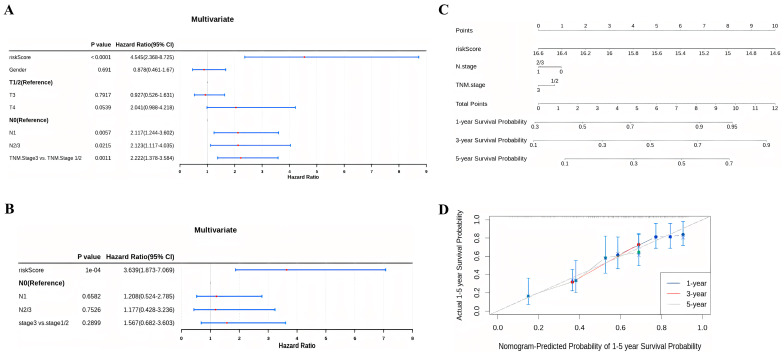
Construction of a nomogram for OS prediction in ESCC. (**A**,**B**) Univariate (**A**) and multivariate (**B**) Cox regression analysis of prognostic clinical indicators. (**C**) Nomogram to predict the 1-year, 3-year, and 5-year OS rates in ESCC patients. (**D**) Calibration curve used to evaluate the accuracy of the nomogram model at 1-year, 3-year, and 5-year time points.

**Figure 5 cancers-15-04399-f005:**
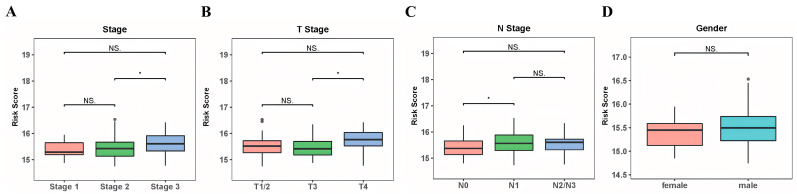
Association between clinicopathological data and risk score in ESCC. (**A**−**D**) The boxplot shows the association between the risk score and clinical stage (**A**), T stage (**B**), N stage (**C**), and gender (**D**). The “•” represents outliers. ^NS^ *p* > 0.05 and * *p* < 0.05.

**Figure 6 cancers-15-04399-f006:**
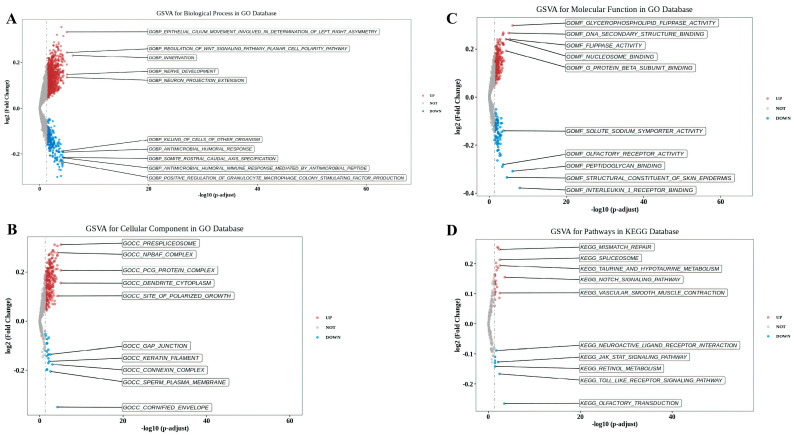
Gene Set Variation Analysis (GSVA) of high-risk and low-risk groups based on the prognostic signature of oxidative stress-related DElncRNAs. (**A**–**C**) Enriched categories of biological process (**A**), cellular component (**B**), and molecular function (**C**) in the high-risk and low-risk groups. (**D**) Significant Kyoto Encyclopedia of Genes and Genomes (KEGG) pathways in the high-risk and low-risk groups.

**Figure 7 cancers-15-04399-f007:**
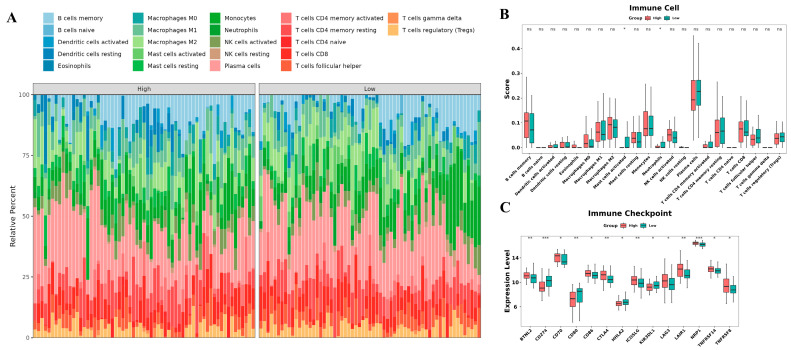
Immune-related analysis using the prognostic signature in ESCC. (**A**) Stacked column chart showing the proportions of tumor-infiltrating cells in the high-risk and low-risk groups. (**B**) Boxplots comparing the immune cell levels between the high-risk and low-risk groups. (**C**) Boxplots comparing the expression levels of immune checkpoint genes between the high-risk and low-risk groups. ^ns^ *p* > 0.05, * *p* < 0.05, ** *p* < 0.01, and *** *p* < 0.001.

**Figure 8 cancers-15-04399-f008:**
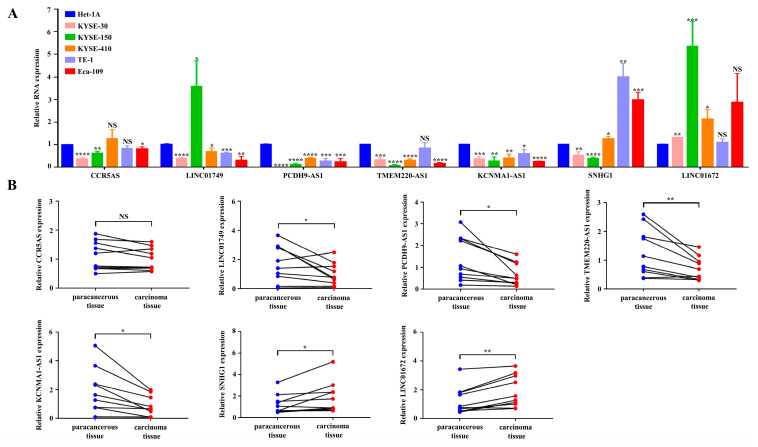
Verification of the expression levels of the seven oxidative stress-related DElncRNAs in cell lines and tissues. (**A**) Relative expression of the seven oxidative stress-related DElncRNAs in the normal human esophageal epithelial line Het-1A and ESCC cells (KYSE-30, KYSE-150, KYSE-410, TE-1, Eca-109). (**B**) Relative expression of the seven oxidative stress-related DElncRNAs in 10 pairs of carcinoma tissue and paracancerous tissue of ESCC. ^NS^
*p* > 0.05, * *p* < 0.05, ** *p* < 0.01, *** *p* < 0.001, and **** *p* < 0.0001.

**Figure 9 cancers-15-04399-f009:**
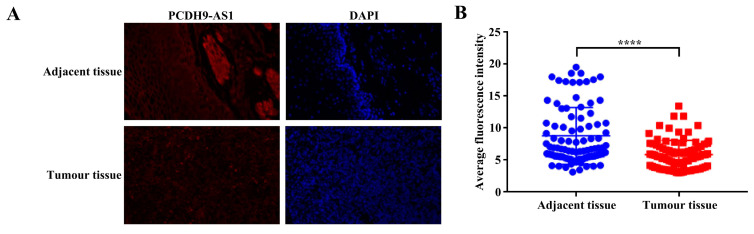
Immunofluorescence analysis of PCDH9-AS1 expression in carcinoma tissues and paracancerous tissues of ESCC. (**A**) Representative images (magnification 200×). DAPI was used for nuclear staining. (**B**) Quantitative map of the relative fluorescence intensity for 80 pairs of carcinoma tissue and paracancerous tissue of ESCC. **** *p* < 0.0001.

**Figure 10 cancers-15-04399-f010:**
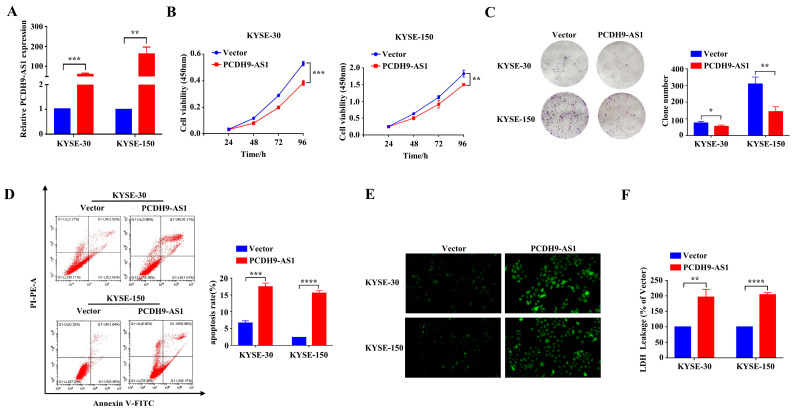
The effect of PCDH9-AS1 overexpression on ESCC cells. (**A**) Relative expression level of PCDH9-AS1 after transfection with the corresponding pcDNA 3.1 (+) vector. (**B**–**D**) Cell proliferation ability (**B**), colony formation ability (**C**), and apoptosis level (**D**) in the PCDH9-AS1 overexpression and control groups. (**E**,**F**) Oxidative stress level evaluated by detecting ROS (**E**) (green, magnification 200×) and LDH (**F**) levels in the PCDH9-AS1 overexpression and control groups. * *p* < 0.05, ** *p* < 0.01, *** *p* < 0.001, and **** *p* < 0.0001.

**Table 1 cancers-15-04399-t001:** Primer sequences.

Targets	Forward Sequence (5′→3′)	Reverse Sequence (5′→3′)
CCR5AS	AACATTTGGTGCCGAAGACC	CATGGAGTGAGGGTGAGGAG
LINC01749	GGCCTCTCTTGAAGGGACTT	GGCCTGACACACGAATGTTT
PCDH9-AS1	TTTAGGAAAGGAACTATTATCAC	GCTTATTATTGCCTATAAACGAC
TMEM220-AS1	AGGGAGCCACTCTGCCCTTGTTT	ATGAGGACTGTGAAGCCGAGAAA
KCNMA1-AS1	F: GGGACATTGGGAGGAACAGA	ACCAGCAGGGCTAATAGCAG
SNHG1	F: CCTGCAAGCCTCTTGCTTAG	TGGGCTGAACATTGCAACAA
LINC01672	F: GGCAAAAACCAGGAGATCCCA	GCCATGTCATTAGCCACCAG
GAPDH	F: ACCCACTCCTCCACCTTTGA	CCACCCTGTTGCTGTAGCCA

**Table 2 cancers-15-04399-t002:** Correlation between PCDH9-AS1 expression and clinicopathological parameters of ESCC patients.

Characteristic	*n*	PCDH9-AS1 Level		*p-*Value
		Low (*n* = 40)	High (*n* = 40)	
Gender				
Male	56	30	26	0.3291
Female	24	10	14	
Age (years) ^#^				
≤60	33	19	14	0.2958
>60	46	21	25	
Histological grade				
I + II	65	34	31	0.3902
III + IV	15	6	9	
Lymph node metastasis				
Negative	41	18	23	0.2634
Positive	39	22	17	
Tumor Stage ^#^				
T1 + T2	9	1	8	0.0714
T3 + T4	27	12	15	
Clinical stage ^#^				
I + II	19	4	15	0.0258
III + IV	28	15	13	

^#^ The totals may not correspond to the sum of the individual numbers due to missing data.

## Data Availability

Any data and R scripts in this study can be obtained from the corresponding author upon reasonable request.

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
