# Peer review of "Development and Validation of the Oxidative Stress Related lncRNAs for Prognosis in Esophageal Squamous Cell Carcinoma"

_cancers, 2023, doi:10.3390/cancers15174399_

Round 1

Reviewer 1 Report

Overall

I congratulate authors of providing nicely conducted study and especially on how visually clearly results have been shown. General motivation for the study plan as it is and introduction to ECC is presented well.

Major issues

-       One of most major concerns for the study are related to the dataset GSE53625

1.     On lines 80-81 authors write about careful selection of data before datamining. However, without actual, explicit and reproducible instructions how data has been selected from GSE53625 research cannot be reproduced nor understood. This needs to be clarified.

2.     Dataset GSE53625 has been used in several studies, which as such is very good practice to fully utilize valuable research materials and resources, however in this case this does raise some concerns. Other studies listed using GSE53625 are obviously ECC related as it is ECC dataset, but their results indicate findings of immune- or lncRNA-related associations to ECC prognostics or functional characterization. Yet, unless I am mistaken, are not even referenced by this study. This raises some concerns why not? Results must be related and either provide additional supportive evidence to understand factors suitable for ECC prognostics or indicate contradictive results. Authors correctly raise point in line 523 about bias from datasource, yet fail to properly position these results into sphere of results generated from the same dataset earlier. This must be corrected. Especially, consider my later point regarding oxidative stress related nature of these seven lncRNAs and prognostic nature of lncRNAs alone.

-       Association of the functions of lncRNAs to oxidative stress is primarily based on correlation with expression of oxidative stress related genes. While useful hypothesis, it is not a causal link.

-       In line 82 authors claim that dataset in question does have comprehensive clinical information, and in line 299 you use word “various” in the context of which clinical variables you used to study the assocations between developed risk factor and clinicopathological traits. If dataset does have comprehensive clinical information, and you used various ones of these, why you list results only from most ordinary ones like TNM stages and gender? What were the other ones which did not have association? Was any of those something that reveals immunosignature strength within tumor tissue?

-       Line 309, entire chapter 3.5. These results are not properly interpreted here nor later in the paper do they actually support primary results of this paper? What does enrichment of these functionals in high- vs low-risk mean for the prognostic value of lncRNAs?

-       Second major concern relates to the claim of these seven lncRNAs being related to oxidative stress. While in vitro experiments do support that at least PCDH9-AS1 having quite clear functional association to ROS amounts etc… I still do not consider link between oxidative stress and these seven lncRNA to be much more than correlation. That being said, it does not reduce potential prognostic value of these lncRNAs, but yet I ask authors to clearly rephrase and reference in more detail if these is actual functional explanation known between these lncRNAs and oxidative stress or is it just accumulating statistical correlation from various studies.

-       This major concern is related to independence of developed prognostic model based on lncRNAs to prognostic models developed for ECC based on immunosignature. Amount of immune related cells and their stage/activity are highly related to oxidative stress, like authors state as well, and these are known to be prognostic factors in many epithelial cancers. Multivariate Cox (Fig 4B) results show that lncRNA model is independent  of TNM based ones, but please see my question above regarding other clinicopathological variables and using those as well. In overall, consider this counter-argument for your study :

“You have found prognostic lncRNAs which are correlatively linked to oxidative stress related genes, which are either same or correlated to genes reporting immunoactivity. Thus, have you found prognostic markers (lncRNAs) for immunosignature and immunoactivity or independent prognostic markers for oxidative stress?”

Think how to mitigate this argument and write it to the paper. Obviously, for some pure prognostic use this wouldn’t even matter, but I need to ask you to clarify this properly. Separate suggestive results from proven ones, and please take into consideration my earlier comments regarding that this dataset has been used to find immunorelated prognostic markers already. So that signal is in the data, make certain to convince all readers that you are just not seeing shadow of this via lncRNA but instead novel one. Or is that it doesn’t even matter because lncRNAs as such provide more convenient/applicable prognostic target that immunorelated gene expression/tissuemarkers.

Minor issues

-       Line 69, there is a typo (ang)

-       In line 271 authors use word “developing” in the context of nomogram. Did you actually develop said plot type or plot type variation as novel, or is it just wrong choice of word and you meant that you did “draw”, “construct” or “calculate etc.. this specific plot

-       In line 424 authors state role of oxidative stress in ECC, please clarify if it is especially high in ECC or maybe not known or so…?

Other issues

-       I highly recommend authors (and journal) to consider the practice of publishing used analysis code (Github repo or folder of code files etc..) as that is actually only way to guarantee reproducibility and proper scientific process of the presented research. 

Reviewer 2 Report

The manuscript provides a comprehensive overview of the critical challenges posed by esophageal cancer (EC) globally, particularly focusing on its prevalence, pathology, and treatment limitations. The description of EC's subtypes and their predominance sets the stage for understanding the disease's complexity.

The manuscript effectively introduces the role of long non-coding RNAs (lncRNAs) in the context of cancer biology. The connection between aberrant lncRNA expression and their contribution to various aspects of malignancy is well articulated, underlining their potential as diagnostic and prognostic biomarkers.

 The proposed study's methodology is outlined effectively, focusing on the identification of differentially expressed oxidative stress-related lncRNAs in ESCC patients. The subsequent construction of a prognostic risk model utilizing these lncRNAs is a logical progression, promising valuable insights into prognostic assessment for patients with ESCC.

While the manuscript introduces the interplay between oxidative stress and lncRNAs, elaborating on the molecular mechanisms through which lncRNAs modulate oxidative stress or are influenced by it would enhance the depth of understanding.

To underscore the significance of the proposed research, elaborating on the potential impact of the prognostic risk model on clinical practice, patient outcomes, and treatment decisions could be beneficial.

Addressing the points mentioned above would further strengthen the manuscript and provide a comprehensive understanding of the study's significance and potential implications.

Author Response

Response to Reviewer 2 Comments

Point 1: The manuscript provides a comprehensive overview of the critical challenges posed by esophageal cancer (EC) globally, particularly focusing on its prevalence, pathology, and treatment limitations. The description of EC's subtypes and their predominance sets the stage for understanding the disease's complexity.

The manuscript effectively introduces the role of long non-coding RNAs (lncRNAs) in the context of cancer biology. The connection between aberrant lncRNA expression and their contribution to various aspects of malignancy is well articulated, underlining their potential as diagnostic and prognostic biomarkers.

The proposed study's methodology is outlined effectively, focusing on the identification of differentially expressed oxidative stress-related lncRNAs in ESCC patients. The subsequent construction of a prognostic risk model utilizing these lncRNAs is a logical progression, promising valuable insights into prognostic assessment for patients with ESCC.

While the manuscript introduces the interplay between oxidative stress and lncRNAs, elaborating on the molecular mechanisms through which lncRNAs modulate oxidative stress or are influenced by it would enhance the depth of understanding.

Response 1: Thank you for your feedback. We updated the discussion section to include your suggestion about the molecular mechanisms for the interplay between oxidative stress and lncRNAs.

The specific content is as follows:

“Additionally, Liu et al. constructed a prognostic model pertaining to oxidative stress in ESCC based on the integration of data from the TCGA, GTEx, and GEO databases. This model aims to predict the prognosis of ESCC patients and carries considerable clinical significance[37]. Consequently, the development of oxidative stress-related biomarkers is crucial for predicting the prognosis of ESCC patients. Owing to their diverse array and intricate spatial configurations, lncRNAs possess the capacity to regulate expression at multiple levels, including transcription and translation [38]. Numerous studies have demonstrated, under long-term or short-term oxidative stress, the expression levels of lncRNAs are dysregulated. For example, in gastric cancer, DNA damage response induced by oxidative stress can promote the binding of H3K27ac and CREBBP, thus facilitating the expression of lncRNA NORAD [39]. In cholangiocarcinoma, oxidative stress can also upregulate the expression of lncRNA H19 and lncRNA HULC [40]. On the other hand, lncRNAs can affect the levels of oxidative stress in tumor cells through various mechanisms. For instance, in hepatocellular carcinoma, LINC01134 recruits the transcription factor SP1 to the p62 promoter to activate the antioxidant pathway of p62 [41]; lncRNA GABPB1-AS1 interacts with GABPB1 to inhibit its translation, leading to decreased PRDX5 expression and ultimately impaired antioxidant capacity of cells [42]. Additionally, the crosstalk between lncRNAs and oxidative stress can modulate various cancer-related signaling pathways such as p53, NF-κB, Nrf2, AKT, EGFR, FOXO3, Keap1, PTEN and Wnt [43,44].”

Point 2: To underscore the significance of the proposed research, elaborating on the potential impact of the prognostic risk model on clinical practice, patient outcomes, and treatment decisions could be beneficial.

Response 2: Thank you for your feedback. We have added this description in the discussion section of the manuscript as follows:

“In this study, we constructed a prognostic model based on seven lncRNAs associated with ESCC patient prognosis. By computing the patients' risk scores, we predict their prognoses, wherein patients in the high-risk group exhibit poorer outcomes. Prognostic models hold the potential to anticipate the trajectory of patients' future developments, enhancing comprehension of disease progression. Furthermore, we performed an independent prognostic analysis and developed a nomogram model grounded in the risk score and clinical features of the samples. This model transforms risk score, N stage, and TNM stage into scores to predict patient survival rates at 1, 3, and 5 years, offering clinical utility. Similarly, immune infiltration-related analyses also provide valuable insights for incorporating immune therapy into the treatment of ESCC patients and assessing immunotherapeutic efficacy. In summary, the oxidative stress-related prognostic model we constructed for ESCC patients using bioinformatics methods presents novel directions and targets for ESCC treatment and prognosis enhancement. However, the effectiveness and clinical applicability of this model necessitate further validation.”

Point 3: Addressing the points mentioned above would further strengthen the manuscript and provide a comprehensive understanding of the study's significance and potential implications.

Response 3: Thank you for the comments. Based on your suggestions, we have modified the Discussion section of the manuscript as follows:

“This study aimed to identify seven DElncRNAs (CCR5AS, LINC01749, PCDH9-AS1, TMEM220-AS1, KCNMA1-AS1, SNHG1, and LINC01672) associated with oxidative stress and develop a risk model and nomogram for accurate prognosis prediction in patients with ESCC, which offers novel approaches for predicting and improving the prognosis of ESCC patients, thereby providing new avenues for ESCC treatment. Additionally, the study investigated the relationship between the risk model and the immune environment. Immune-related analyses, particularly those involving immune checkpoints, furnish theoretical support for the application of immune therapy. Concurrently, novel insights were gained into the role and potential mechanisms of a specific long non-coding RNA, PCDH9-AS1, in regulating the development of ESCC. Our research findings establish a new theoretical foundation for exploring the roles of lncRNAs and oxidative stress in ESCC, presenting fresh targets for ESCC treatment. Moreover, we will persist in investigating the roles of these prognostically relevant lncRNAs, delving into their potential molecular mechanisms. As our study unfolds, we anticipate gaining a deeper understanding of these mechanisms and their implications. In conclusion, these findings have important implications for prognosis prediction, diagnosis, and treatment of esophageal squamous cell carcinoma in clinical practice.”

Round 2

Reviewer 1 Report

Dear authors,

Thank you for detailed and clear response letter. I do not find any other points of concern and therefore recommend publishing manuscript in its current form.

Reviewer 2 Report

The Authors have answered all the queries. The manuscript can be accepted.